# The Significance of FilmArray Blood Culture Identification Panel (FA-BCID) for Managing Patients with Positive Blood Cultures

**DOI:** 10.3390/diagnostics13213335

**Published:** 2023-10-29

**Authors:** Kristin Widyasari, Seungjun Lee, Oh-Hyun Cho, Sun-In Hong, Byung-Han Ryu, Sunjoo Kim

**Affiliations:** 1Institute of Medical Science, Gyeongsang National University, Jinju 52828, Republic of Korea; kristinwidyasari@gmail.com; 2Department of Laboratory Medicine, Gyeongsang National University Changwon Hospital, Changwon 51472, Republic of Korea; sjlee0318@gmail.com; 3Department of Laboratory Medicine, College of Medicine, Gyeongsang National University, Jinju 52828, Republic of Korea; 4Department of Internal Medicine, Soonchunhyang University Cheonan Hospital, Cheonan 31151, Republic of Korea; zenmd@naver.com; 5Division of Infectious Diseases, College of Medicine, Soonchunhyang University Cheonan Hospital, Soonchunhyang University, Cheonan 31151, Republic of Korea; hsun0702@hanmail.net; 6Department of Internal Medicine, Anyang SAM Hospital, Anyang 14030, Republic of Korea; qudhany@gnuh.co.kr

**Keywords:** blood culture, pathogen identification, antimicrobial stewardship program, FA-BCID, MALDI-TOF

## Abstract

We analyzed the accuracy and time efficiency of the FilmArray blood culture identification (FA-BCID) panel in identifying the pathogens in positive blood cultures. Two-hundred and seventy-two individuals were randomly assigned as the control (*n* = 212) and FA-BCID (*n* = 60) groups participating in this study. Matrix-assisted laser desorption/ionization time-of-flight mass spectrometry (MALDI-TOF MS) was used to assess the control group. Meanwhile, the FA-BCID group was evaluated using both FA-BCID and MALDI-TOF, and the results were compared. The identification results from 73% (44/60) of the blood samples demonstrated agreement between FA-BCID and MALDI-TOF. The FA-BCID panel detected *mecA* genes in seven *Staphylococcus* species; six cases were confirmed using antimicrobial susceptibility testing. In addition, *KPC* genes were detected in one *Escherichia coli* and one *Klebsiella pneumoniae*, although only the latter corresponded with the result from antimicrobial susceptibility testing. The turnaround time (TAT) for identification through FA-BCID was shorter, with a median of 3.6 [2.4–4.6] hours (*p* < 0.05). No significant differences in the clinical and microbial outcomes following the ASP were observed between FA-BCID and MALDI-TOF. These results suggest that the FA-BCID panel provides an identification result that is as reliable as that provided by the routine identification procedure but with shorter TAT; thus, the FA-BCID method is considered an effective and beneficial method for therapeutic decision making and the improvement of the ASP for patients with bloodstream infection.

## 1. Introduction

Sepsis is a life-threatening organ dysfunction caused by a dysregulated host response to infection [1]. It causes significant morbidity and mortality in hospitals [2] and is a significant health and financial burden in many countries. Sepsis kills 11 million people annually and disables millions more [3]. In Korea, the burden of sepsis is also considered high and is expected to increase, considering the aging population [4]. The estimated mortality rate due to sepsis in Korea from 2012 to 2022 was around 30% [5,6]. The onset of proper antimicrobial medication and the duration of infection are substantially correlated with survival [7]. An accurate and immediate diagnosis enables appropriate treatment, improving patient outcomes, shortening intensive care stays, reducing health costs, and lowering the mortality rate.

Since sepsis is related to the infection of bacteria, fungi, or other pathogens, detecting and identifying the related pathogen in the blood is crucial for treating sepsis patients. Among diagnosis methods, blood culture is considered the gold standard for detecting that bacteria or fungi that cause sepsis in the blood [8]. Nevertheless, conventional methods for identifying microorganisms in blood cultures, such as agar-based techniques, require approximately 72 h [9]. Other methods routinely used as an identification method, such as manual biochemical tests, matrix-assisted laser desorption ionization time-of-flight mass spectrometry (MALDI-TOF MS), or automated identification systems such as VITEK^®^ 2 (bioMérieux SA, Marcy-l’Étoile, France), also require approximately two to three days before the results are available [10]. Moreover, these methods cannot detect antimicrobial resistance genes in the identified microorganisms. Therefore, another test for antimicrobial susceptibility tests (ASTs), such as Kirby–Bauer disk diffusion, an Epsilometer test (*E* test, bioMérieux), or automated AST systems such as VITEK^®^ 2, are generally required. The additional assessment in conventional blood culture identification undoubtedly adds to the length of time in which the results are available [9]. In addition, traditional methods are relatively labor intensive and require expertise for evaluation; thus, alternative identification methods that are somewhat less labor intensive yet enable rapid detection are preferable for clinical diagnosis [8,9].

The FilmArray blood culture identification panel (FA-BCID; BioFire, Salt Lake City, UT, USA) is a closed diagnostic system using high-order multiplex PCR analysis and is approved by the U.S. Food and Drug Administration (FDA) for use on positive blood cultures [11]. FA-BCID enables a simultaneous test with an automated readout of results directly from a positive blood culture within one hour [9]. This system allows for rapid detection of pathogens and antibiotic resistance genes associated with bloodstream infection, with 27 total targets comprising 24 pathogens (19 bacteria, nine yeasts) and 3 antibiotic resistance genes (*mecA*, *yanA/B*, and *blaKPC*) [12]. Several studies have reported the accuracy of FA-BCID for identifying microorganisms in positive blood cultures [13,14]; however, the actual clinical implementation of FA-BCID is still considered to be low.

In this study, we evaluated the clinical benefit of the FA-BCID panel by assessing its accuracy in prospective clinical samples and determining its turnaround time compared to routine identification methods. We also defined the benefits of FA-BCID when implemented in the antimicrobial stewardship program (ASP) by evaluating the clinical and microbial outcomes compared to the conventional method.

## 2. Methods

### 2.1. Study Design

We conducted a prospective study at the Gyeongsang National University Changwon Hospital, Republic of Korea. Two-hundred and seventy-two positive blood cultures were collected between November 2018 and April 2019 and included as study subjects. The study subjects were randomly assigned to the control (*n* = 212) and FA-BCID (*n* = 60) groups. Samples obtained from patients admitted to the emergency department in GNUCH that were suspected of a positive blood culture were included in the study. The samples from patients who died between blood culture sampling and positivity detection were excluded from the study.

The experiment was conducted during the same period as sample collection (November 2018 and April 2019). Only individuals who conducted investigation and data analysis have access to information that could identify individual participants during and after data collection. The identification of microorganisms in the positive blood cultures was conducted using the FA-BCID panel and MALDI-TOF (Figure 1). This study was approved by the Institutional Review Board of Gyeongsang National University Changwon Hospital (IRB No. 2018-07-15). Written informed consent was waived because of the retrospective nature of the study and the analysis used anonymous clinical data.

### 2.2. Laboratory Test

#### 2.2.1. Blood Culture

According to the manufacturer’s instructions, the blood cultures were processed using BACT/ALERT Virtuo (bioMérieux, Marcy-l’Étoile, France) [15]. Two blood culture bottles (FA plus and FN plus) were prepared for each episode and loaded onto the motion-activated loading bay. Subsequently, the bottles were automatically moved to a carousel and into the intelligent scanning station. Further, the blood fill level in the bottles was sensed, recorded in real time, and subjected to detection. The bottles were removed from the system when a positive signal was detected. Then, a small amount of broth from the positive blood culture was sampled and spread on suitable agar plates or directly subjected to identification using FC-BCID. The microorganisms grown on the agar plates were subjected to Gram staining and conventional identification using MALDI-TOF.

#### 2.2.2. FilmArray Blood Culture Identification (FA-BCID)

FA-BCID combines nucleic acid extraction, high-order nested multiplex PCR, and post-PCR DNA melting curve analysis within one system. The assessment was conducted according to the manufacturer’s instructions and as described previously [9]. Briefly, 100 µL of the broth from the positive blood culture bottle was diluted in 500 µL sample dilution buffer; then, 300 µL from this mixture was injected into the FA pouch for analysis. Nucleic acid extraction, amplification, detection, and analysis were wholly automated within the FilmArray system.

#### 2.2.3. MALDI-TOF MS

Following Gram staining, microorganisms from the positive blood cultures were sub-cultured on the proper agar plate (MacConkey agar plate for Gram-negative bacteria and blood agar plate for Gram-positive bacteria). The microorganisms that grew on the plates were identified using VITEK^®^ MS v3.0 (bioMérieux). This automated mass spectrometry microbial identification system uses matrix-assisted laser desorption ionization time-of-flight (MALDI-TOF) technology [16]. The assessment was conducted following the manufacturer’s instructions. The target slide was briefly prepared and introduced to the high-vacuum environment, where a laser burst ionized the samples, and the proteins’ flight time was recorded. The proteins were detected with a sensor, and the spectrum representing the protein composition of each sample was generated. These protein compositions were used to identify microorganisms that were being assessed.

#### 2.2.4. Antimicrobial Susceptibility Test (AST)

The antimicrobial susceptibility test (AST) was conducted using VITEK^®^ 2 AST Cards (bioMérieux) with pure microorganisms one day after the empirical antimicrobial therapy was implemented. The AST was performed according to the manufacturer’s instructions. Briefly, for each isolate, the suspensions were prepared and used to inoculate the AST card. The results were read and interpreted automatically using software (VITEK^®^ 2 Advanced Expert System, version 9.02) provided by bioMérieux [17].

#### 2.2.5. Analysis of the Clinical and Microbiological Outcomes

We determined several clinical and microbial factors to evaluate the FA-BCID outcomes of the antimicrobial stewardship program (ASP). We observed their outcomes following the ASP that was performed based on the FA-BCID result. Subsequently, the results were compared with those from the control group, in which the implementation of the ASP was based on the MALDI-TOF result. Three infectious disease physicians reviewed the medical records and consulted regarding escalating/de-escalating antibiotics usage.

### 2.3. Statistical Analysis

All statistical analyses were performed using MedCalc for Windows, version 18.5 (MedCalc Software, Ostend, Belgium), and *p* < 0.05 was considered significant. The two groups were compared using the *t*-test for continuous variables and the chi-squared test for categorical variables

## 3. Results

### 3.1. Study Participants 

We included a total of 272 positive blood cultures from 272 unique patients and randomly distributed them into control (*n* = 212) and FA-BCID (*n* = 60) groups. Analysis of participants’ demographic and clinical characteristics demonstrated no significant differences between the two groups (Table 1).

### 3.2. The Diversity of Clinical Isolates in the Blood Culture Detected Using MALDI-TOF and FA-BCID

We assessed and compared the diversity of the clinical isolates from the 60 positive blood samples in the FA-BCID group using MALDI-TOF and FA-BCID methods. From 60 positive blood samples, about 67 isolates were identified using the MALDI-TOF method; meanwhile, 76 isolates were identified using the FA-BCID method (Appendix A). The microorganism identification results from 73% (44/60) of the blood samples demonstrated agreement between FA-BCID and MALDI-TOF groups. One yeast strain, *Candida tropicalis*, was identified in three positive blood samples via MALDI-TOF and FA-BCID. Five positive blood samples showed polymicrobial growth following the assessment with both MALDI-TOF and FA-BCID, and three positive blood samples showed polymicrobial increase only in the MALDI-TOF assessment. Nine positive blood samples exhibited polymicrobial growth only in the FA-BCID assessment, and forty-three positive blood samples showed monomicrobial growth following evaluation with both MALDI-TOF and FA-BCID. Among the identified isolates from the samples with monomicrobial growth, three isolates (5%) were differentially identified using the FA-BCID panel (not placed, *Staphylococcus aureus*, and *Proteus*) and MALDI-TOF (*Bacteroides fragilis*, *Staphylococcus caprae*, and *Citrobacter younger*), respectively. The FA-BCID panel was used to detect *mecA* genes in seven *Staphylococcus* species; six were confirmed with antimicrobial susceptibility testing. The *KPC* genes were also detected in one *Escherichia coli* and one *Klebsiella pneumoniae*, although only the latter agreed with antimicrobial susceptibility testing (Appendix A).

### 3.3. Microorganisms Identified from the Positive Blood Samples According to the Target in the FA-BCID Panel

The FA-BCID panel includes 27 target microorganisms and three antibiotic resistance markers (Table 2). This study demonstrated that a total of 64 organisms that were identified from the positive blood samples by routine procedure (MALDI-TOF) are in concordance with the target organisms in the FA-BCID panel. Among these organisms, 63 were identified using MALDI-TOF and FA-BCID, and 1 isolate of *Enterococcus* spp. was only detected using MALDI-TOF. Three isolates, i.e., *Shewanella algae*, *Clostridium tertium*, and *Bacteroides fragilis*, were only detected in the samples assessed using MALDI-TOF. We also observed 14 isolates detected only when the assessment was conducted using FA-BCID but not MALDI-TOF. Among these 14 isolates, eight isolates were identified as *Proteus* spp.

### 3.4. Clinical Efficacy of FA-BCID Detection for the Antimicrobial Stewardship Program

In this study, we observed a shorter time for identifying microorganism pathogens in blood culture using FA-BCID compared to MALDI-TOF (control). The median turnaround time (TAT) in the FA-BCID group was 3.6 h, whereas the TAT in the control group was 28.5 h (*p* < 0.0001) (Figure 2). Subsequently, the identification results were reported to the attending physician and infectious disease physicians; thus, immediate action, such as the ASP, could be carried out to treat the patients.

We also observed that the time for escalation of the initial antibiotic appeared to be significantly shorter in the FA-BCID group (21.9 h) than in the control group (50.2 h) (*p* = 0.0274). Correspondingly, the time for the antimicrobial susceptibility test (AST) and the de-escalation of antibiotics was also shorter in the FA-BCID group than in the control (Figure 2).

Furthermore, observation of the clinical and microbial outcomes in the patients following the ASP demonstrated a comparable result between the FA-BCID and control groups. The clinical outcomes, including disposition, length of stay, and mortality, were not statistically different between the two groups (Table 3). The number of patients who recovered after a blood infection was 75% in the FA-BCID group and 74.5% in the control group. The average length of stay (entire hospitalization stay) was 18 days in the FA-BCID group and 15 days in the control (*p =* 0.7201). There were also no significant differences in 30-day mortality (16.7% vs. 14.2%, *p* = 0.6278) and 30-day readmission for infection of the same organism (1.7% vs. 0.5%, *p* = 0.3325) between the FA-BCID and control groups. Similarly, we did not observe any statistical difference in microbiologic outcomes between the two groups. The acquisition of *C. difficile* or multidrug-resistant organisms appeared comparable for both groups, with a *p*-value of 0.7503.

## 4. Discussion

In this study, we presented a prospective study to evaluate the advantage of microorganism identification in blood cultures using FA-BCID, a multiplex PCR-based system. According to the published research, FA-BCID demonstrated high sensitivity and specificity for pathogen identification: up to 96.5% (95% CI, 91.3–99.0) and 99.7% (95% CI, 99.3–99.9), respectively [18]. The target microorganisms available in the FA-BCID panels are the most commonly encountered bacterial and fungal organisms that cause bloodstream infection, including *Escherichia coli*, *Klebsiella pneumoniae*, *Klebsiella oxytoca*, *Enterobacter cloacae*, *Proteus* sp., *Pseudomonas aeruginosa*, *Serratia marcescens*, *Haemophilus influenza*, *Neisseria meningitides*, *Candida albicans*, *Candida glabrata*, *Candida parapsilosis*, *Candida krusei*, and *Candida tropicalis* [11,19].

The result from this study demonstrated that the FA-BCID panel was 97% (58/60) reliable when used to detect the presence of microorganisms in positive blood samples; 77% (46/60) of these microorganisms demonstrated monomicrobial growth. This suggests that the FA-BCID panel could be used to cover the majority of microorganisms encountered in the positive blood cultures during the study period. The results of this study are consistent with a previous study that reported 91% of all assessed cases could be identified with FA-BCID [20].

In this study, microorganisms identified in 73% (44/60) of positive blood cultures demonstrated an agreement between FA-BCID and MALDI-TOF. Microorganisms from six positive blood cultures that showed discrepancy were identified as polymicrobial using FA-BCID and monomicrobial using MALDI-TOF. The presence of *Proteus* sp. in the FA-BCID assessment is the main reason for this discrepancy. As already informed by bioMérieux, the manufacturer of FA-BCID, *Proteus* sp. was likely detected in the blood culture sample due to the increased level of nucleic acid from nonviable *Proteus* sp. bacteria in the BACT/ALERT^®^ blood culture bottles [21]. This condition may increase the risk of false positive *Proteus* sp. using the FA-BCID panel and the BACT/ALERT^®^ blood culture bottles. Additionally, there were three other positive blood cultures with discrepancies between FA-BCID and MALDI-TOF in their identified microorganism, likely due to the limited number of target organisms in FA-BCID, as it only covered 24 different target organisms. The *Shewanella algae*, *Clostridium tertium*, and *Bacteroides fragilis* detected using MALDI-TOF are not target organisms in the FA-BCID panel. The limited number of target organisms in the FA-BCID panel may constitute a disadvantage of FA-BCID implementation compared to the conventional procedure that enables the detection of limitless organisms. However, unlike the traditional procedure, which requires a separate assessment for the detection of antimicrobial resistance genes, FA-BCID is used not only to detect and identify the microorganism, but also to distinguish the isolates with antimicrobial resistance genes [11,19,22], making the diagnosis procedure less complicated and less time-consuming.

Seven of the nine identified isolates, six identified as *Staphylococcus* sp. and one as *Klebsiella pneumoniae*, showed the presence of antimicrobial resistance genes; these were identified as *mecA* and *KPC*, respectively. An antimicrobial susceptibility test confirmed that all six *Staphylococcus* sp. isolates were resistant to methicillin. Similarly, the *Klebsiella pneumoniae* isolate identified as carrying *KPC* was resistant to a variety of *β*-lactam and non-*β*-lactam antibiotics, as reported in a previous study [23]. Identifying these antimicrobial resistance genes would be clinically meaningful if an appropriate antibiotic was escalated. The ability to identify and detect bacteria with antimicrobial resistance genes makes the FA-BCID an adequate clinical diagnostic tool that is beneficial in assisting the physician in providing appropriate treatment for patients with a blood infection. However, given the distribution of resistant bacteria in Korea, a detection panel that also enables one to detect extended-spectrum β-lactamase (*ESBL*) or *NDM* resistance genes is essential; thus, the target antimicrobial resistance in the FA-BCID panel, which is limited to *KPC*, *mecA*, and *vanA/B*, may reduce the clinical benefit and implementation of FA-BCID in hospitals in Korea.

The timing of antibiotic therapy is considered a crucial determinant for the survival of patients with a blood infection or sepsis. The mortality rate may be double or more in patients with severe infections who experience delays in the administration of antibiotic therapy [24]. Therefore, a significantly shorter turnaround time of FA-BCID in detecting the pathogens from the positive blood culture may help to prevent a delay in applying an appropriate antibiotic regimen. In this study, the escalation of empiric antibiotics therapy following identification was 2.3 times faster in the FA-BCID group than in the control group. Hence, identifying pathogens from the positive blood culture with FA-BCID may be more efficient as an ASP tool than the conventional methods. Proper and prompt empiric antibiotic therapy for patients with a blood infection significantly reduced the risk of disease complications and mortality [25]. This finding concords with a previously reported study, which also demonstrated the benefit of using FA-BCID for clinical decisions regarding cohort, therapy, and the optimization of the antimicrobial regimen in patients with bloodstream infection [14].

The usage of broad-spectrum antibiotics may contribute to antimicrobial resistance. Hence, reassessing the possibility of any antibiotic resistance following the escalation of antimicrobial therapy is necessary for optimizing the ASP. The de-escalation of antimicrobial therapy is conducted, based on microbiological assessment results, around the third day from the escalation point [26]. Adjusting antimicrobial therapy through de-escalation is crucial to prevent unnecessary antibiotic exposure. Moreover, reducing the number of administered antibiotics may reduce therapy’s side effects and cost. In this study, the de-escalation time of empiric antibiotic treatment in the FA-BCID group was implemented earlier than in the control group. However, unlike antibiotic escalation, which demonstrated a significantly different implementation time, we found that antibiotic de-escalation time between the two groups was similar. We assumed that this condition occurred due to the presence of patients in the FA-BCID group whose condition may have worsened. Although de-escalation of antibiotics is a potential strategy to conserve the effectiveness of broad-spectrum antibiotics in patients who develop a critical condition, a slower antibiotic de-escalation may help maintain protection against bacterial pathogens. Hence, many attending physicians will allow a relatively long time for monitoring before deciding to conduct modification of the administrated antibiotic in patients who develop a critical condition. Nevertheless, as reported by many studies, antibiotic de-escalation in the ASP is considered crucial for minimizing the emergence of multidrug-resistant pathogens [27,28,29]; hence, proper de-escalation by viewing the individual patient, the circumstance, and the overall situation is crucial in the implementation of the ASP.

Considering the rapidity and accuracy of FA-BCID in detecting microbial pathogens in positive blood cultures, we expected that implementing FA-BCID may improve the clinical outcomes of patients. However, this study found no significant differences in the clinical outcomes observed between the two groups. The percentage of clinical outcomes, such as recovery period, length of stays, and mortality rates, were very much alike between the FA-BCID and control groups. These findings show that using FA-BCID may not be advantageous for managing patients with blood infections. We assumed that the relatively small sample size in the FA-BCID group and the unequal distribution of patients with chronic and nonchronic diseases were the main reasons for this finding. Additionally, although FA-BCID may provide rapid and accurate identification of microbial pathogens and detection of antimicrobial resistance genes, the clinical value of FA-BCID also depends on the actual application of the identification results on managing the patients. Hence, effective communication between the diagnosis department and attending physician is crucial to improving the management of patients with blood infections.

The relatively small number of study subjects in the FA-BCID group may constitute a limitation of this study. Additionally, FA-BCID may be less favorable as a bloodborne pathogen detection method due to its high cost and limited targets compared to the conventional method. Nevertheless, as a microbial pathogen detection tool, the FA-BCID panel demonstrated rapid and accurate detection and identification of microbial pathogens and antimicrobial resistance genes; thus, it may constitute a promising AST tool that can be implemented, along with the ASP, to improve management of patients with blood infections.

## Figures and Tables

**Figure 1 diagnostics-13-03335-f001:**
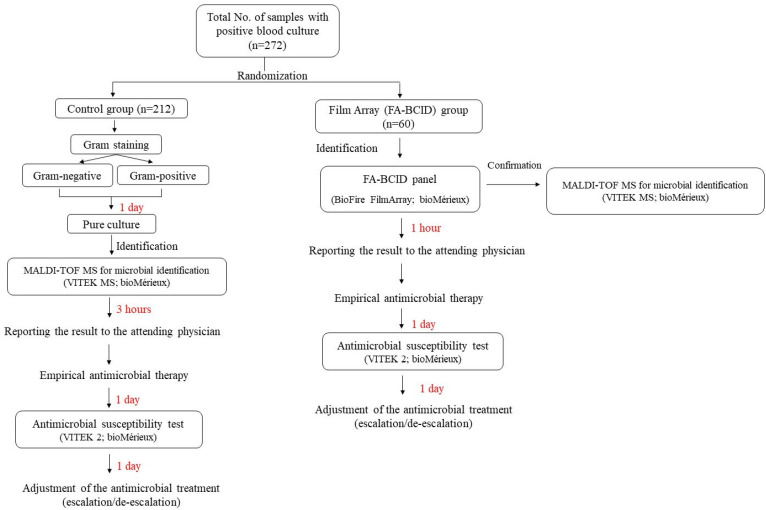
Diagram representing the study design. Two-hundred and seventy-two positive blood culture samples were randomly assigned into control (*n* = 212) and FA-BCID (*n* = 60) groups. MALDI-TOF was used to identify the microorganisms present in the blood in the control group. Meanwhile, in the FA-BCID group, the identification was conducted using both MALDI-TOF and FA-BCID panels.

**Figure 2 diagnostics-13-03335-f002:**
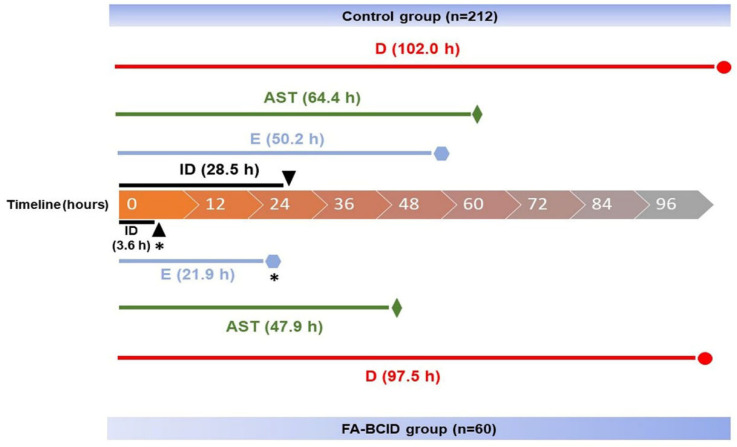
Timeline of pathogen identification, antimicrobial susceptibility test, escalation, and de-escalation of antimicrobial therapy in study groups. Time 0 indicates a report of Gram stain result. Median time in hours (interquartile range [IQR]) to pathogen identification in the control group and FA-BCID group: 28.5 (22.5–41.5) vs. 3.6 (2.4–4.6); escalation: 50.2 (21.5–73.8) vs. 21.9 (5.0–48.6); antimicrobial susceptibility test: 64.4 (46.1–89.9) vs. 47.9 (46.2–94.5); de-escalation: 102.0 (66.9–155.1) vs. 97.5 (53.3–165.0). Asterisk (*) indicates a significant difference between the FA-BCID and control groups with a *p*-value < 0.05. Abbreviations: ID, identification; E, escalation; AST, antimicrobial susceptibility test; D, de-escalation.

**Table 1 diagnostics-13-03335-t001:** Essential demographic and clinical characteristics of study populations.

	Control Group (*n* = 212)	FA-BCID Group (*n* = 60)	*p*-Value
Demographics			
Male proportion (*n*, %)	116 (54.7)	35 (58.3)	0.6194
Age, mean ± SD	69.6 ± 15.0	68.4 ± 14.2	0.5805
Location at enrollment (*n*, %)			0.3623
Outpatient clinic	1 (0.5)	0 (0.0)	
General wards	53 (25.0)	20 (33.3)	
Emergency department	135 (63.7)	37 (61.7)	
ICU	23 (10.8)	3 (5.0)	
Comorbidities, (*n*, %)			
Charlson comorbidity score, mean ± SD	4.9 ± 2.6	4.4 ± 1.9	0.1071
Diabetes mellitus	73 (34.4)	25 (41.7)	0.3038
Hypertension	99 (46.7)	21 (35.0)	0.1078
Cardiovascular disease	35 (16.5)	6 (10.2)	0.2302
Cerebrovascular disease	35 (16.5)	9 (15.0)	0.7795
Chronic heart disease	49 (23.1)	14 (23.3)	0.9716
Chronic liver disease	17 (8.0)	4 (6.7)	0.7295
Chronic lung disease	19 (9.0)	3 (5.0)	0.3212
Chronic renal disease	32 (15.1)	7 (11.7)	0.5044
Malignant tumor	54 (25.5)	15 (25.0)	0.9410
Renal replacement therapy	15 (7.1)	3 (5.0)	0.5688
Neutropenia	4 (1.9)	4 (6.7)	0.0535
Chemotherapy in the prior 30 days	15 (7.1)	6 (10.0)	0.4545
An immunosuppressant in the prior 30 days	1 (0.5)	0 (0.0)	0.5947
Surgery in the prior 30 days	42 (19.8)	6 (10.0)	0.0790
ICU admission	44 (20.8)	14 (23.3)	0.6674
Bacteremia at the time of ICU stay	41 (19.3)	10 (16.7)	0.6402
The severity of illness (*n*, %)			
Pitt bacteremia score, mean ± SD	1.4 ± 2.2	1.5 ± 1.7	0.8810
Requiring mechanical ventilation	28 (13.2)	8 (13.3)	0.9798
Septic shock	49 (23.1)	8 (13.3)	0.1010

Data are presented as No. (%) unless otherwise specified. Abbreviations: FA, FilmArray; SD, standard deviation; ICU, intensive care unit.

**Table 2 diagnostics-13-03335-t002:** Summary of the identified microorganisms and antibiotic resistance genes from 60 samples.

Category	Target ^a^	MALDI-TOF Positive and FA-BCID Positive (*n*)	MALDI-TOF Positive and FA-BCID Negative (*n*)	MALDI-TOF Negative and FA-BCID Positive (*n*)
Gram-negative bacteria	*Acinetobacter baumannii*			
	*Enterobacteriaceae*			
	*Enterobacter cloacae* complex			
	*Escherichia coli*	26		1
	*Klebsiella oxytoca*	2		
	*Klebsiella pneumoniae*	8		
	*Proteus* spp.			8 *
	*Serratia marcescens*			
	*Enterobacteriaceae*, except for the above six targets	2		
	*Haemophilus influenza*			
	*Neisseria meningitidis*			
	*Pseudomonas aeruginosa*	1		1
Gram-positive bacteria	*Enterococcus* spp.	6	1	2
	*Listeria monocytogenes*			
	*Staphylococcus* spp.			
	*Staphylococcus aureus*	7		1
	*Staphylococcus* spp. except *S. aureus*	2		
	*Streptococcus* spp.			
	*Streptococcus agalactiae*	2		
	*Streptococcus pneumoniae*	1		
	*Streptococcus pyogenes*			
	*Streptococcus* spp. except for the above three targets	3		
Yeasts	*Candida albicans*			
	*Candida glabrata*			
	*Candida krusei*			
	*Candida parapsilosis*			
	*Candida tropicalis*	3		
Antimicrobial resistance genes	*KPC*	1		1
	*mecA*	7		
	*vanA/vanB*			
Organisms not included in the FA BCID panel.			3 **	

^a^ FilmArray BCID panel targets organisms. * Manufacturer announced that FA-BCID panel can show false positivity of *Proteus* spp. ** *Shewanella algae*, *Clostridium tertium*, *Bacteroides fragilis* Abbreviations: MALDI-TOF, matrix-assisted laser desorption ionization time-of-flight; FA, FilmArray; BCID, blood culture identification.

**Table 3 diagnostics-13-03335-t003:** Clinical and microbiologic outcomes in the control and FA-BCID groups.

	Control Group (*n* = 212)	FA-BCID Group (*n* = 60)	*p*-Value
Clinical outcomes			
Disposition (*n*, %)			0.1258
Recovery	158 (74.5)	45 (75.0)	
Hopeless discharge	18 (8.5)	1 (1.7)	
Death	36 (17.0)	14 (23.3)	
Length of stay (entire hospitalization) (day, median (IQR))	15 (9–32.75)	18 (8–43)	0.7201
Length of stay (after enrollment) (day, median (IQR))	11 (6–21)	14 (6–31.25)	0.3576
30-day mortality (*n*, %)	30 (14.2)	10 (16.7)	0.6278
30-day attributable mortality (*n*, %)	21 (9.9)	6 (10.0)	0.9828
30-day readmission for infection with the same organism (*n*, %)	1 (0.5)	1 (1.7)	0.3325
Microbiological outcome			
Acquisition of *C. difficile* or multidrug-resistant organisms within 30 days after enrollment	52 (24.6)	16 (26.7)	0.7503

## Data Availability

Data of this study can be available upon reasonable request.

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
