# Peer review of "The Significance of FilmArray Blood Culture Identification Panel (FA-BCID) for Managing Patients with Positive Blood Cultures"

_diagnostics, 2023, doi:10.3390/diagnostics13213335_

Round 1

Reviewer 1 Report

Comments and Suggestions for Authors

Comments and Suggestions for Authors

The manuscript“The significance of FilmArray blood culture identification panel (FA-BCID) in the antimicrobial stewardship program as part of managing patients with positive blood cultures” is relevant and properly conceived and written.

The manuscript is current and relevant, focusing on a subject of scientific interest.

Some details can improve the manuscript so I suggest some minor changes.

Page 2, line 90: The sentence “Written informed consent was waived because of the beneficial addition of FA-BCID to the conventional MALDI-TOF of the study group.” does not make sense. Informed consent is not about whether it is beneficial or not. Or whether the study brings benefits or not. The sentence should be changed so that this aspect is clear. In fact, in my opinion informed consent should have been requested, however it doesn't shock me that they didn't do so, in any case this sentence should be reformulated.

Page 4, line 128: Explain what s/w stands for.

Page 5, line 155: The sentence “From 60 positive blood samples, 67 isolates were identified by MALDI-TOF, and 76 isolates were identified by FA-BCID (Table S1). The microorganism identification results from 73%...” must be explained. Which 60 samples? I supposed the control ones, however it should be more detailed.

Page 8, line 264: Take the italics out.

After these revisions, the paper is able to be published once it is actual, well organized and scientifically relevant.

Author Response

The authors would like to thank the Reviewers for their specific and helpful comments on the manuscript. The authors have carefully considered the comments and have made revisions to the manuscript to address the reviewer’s concerns.

Responses to the Reviewer are reported as follows:

Reviewer 1:

The manuscript“The significance of FilmArray blood culture identification panel (FA-BCID) in the antimicrobial stewardship program as part of managing patients with positive blood cultures” is relevant and properly conceived and written.

The manuscript is current and relevant, focusing on a subject of scientific interest.

Some details can improve the manuscript so I suggest some minor changes.

  1. Page 2, line 90: The sentence “Written informed consent was waived because of the beneficial addition of FA-BCID to the conventional MALDI-TOF of the study group.” does not make sense. Informed consent is not about whether it is beneficial or not. Or whether the study brings benefits or not. The sentence should be changed so that this aspect is clear. In fact, in my opinion informed consent should have been requested, however it doesn't shock me that they didn't do so, in any case this sentence should be reformulated.

Response:

As suggested by the reviewer, we have reformulated the sentence.

Lines: 95-96

“Written informed consent was waived because of the retrospective nature of the study and the analysis used anonymous clinical data.”

  1. Page 4, line 128: Explain what s/w stands for.

Response:

We have corrected the sentence to avoid the misunderstanding.

Line 132-133.

“These protein compositions were used to identify microorganisms that were being assessed.”

  1. Page 5, line 155: The sentence “From 60 positive blood samples, 67 isolates were identified by MALDI-TOF, and 76 isolates were identified by FA-BCID (Table S1). The microorganism identification results from 73%...” must be explained. Which 60 samples? I supposed the control ones, however it should be more detailed.

Response:

As suggested by the reviewer, we have edited the sentence accordingly to give a detailed explanation.

Lines: 163-166.

“We assessed and compared the diversity of the clinical isolates from the 60 positive blood samples in the FA-BCID group by MALDI-TOF and FA-BCID methods. From 60 positive blood, the MALDI-TOF method identified about 67 isolates, meanwhile, the FA-BCID method identified 76 isolates (Table S1).”

  1. Page 8, line 264: Take the italics out.

Response:

           As suggested by the reviewer, we have edited the sentence by taking the italics out.

Line: 274-275

KPC was resistant to a variety of β-lactam and non-β-lactam antibiotics, as reported in a previous study”

After these revisions, the paper is able to be published once it is actual, well organized and scientifically relevant.

Reviewer 2 Report

Comments and Suggestions for Authors

Dear authors

In the current manuscript, you have analyzed the accuracy and time efficiency of the Film Array blood culture identification (FA-BCID) panel in identifying the pathogens in positive blood cultures via testing the results of 272 individuals (212 controls, and 60 FA-BCID participants. Results showed that 73% (44/60) of the blood samples demonstrated agreement between FA-BCID and MALDI-TOF. The FA-BCID panel detected mecA genes in seven Staphylococcus species; six cases were confirmed by antimicrobial susceptibility testing. In addition, KPC genes were detected in one Escherichia coli and one Klebsiella pneumoniae, although only the latter corresponded with the result from antimicrobial susceptibility tests. The turnaround time (TAT) for identification by FA-BCID was shorter. They suggested that the FA-BCID panel provides an identification result that is as reliable as that provided by the routine identification procedure but with shorter TAT; thus, the FA-BCID method is considered an effective and beneficial method for therapeutic decision-making and the improvement of the ASP for patients with bloodstream infection.

The work is very good and interesting. However, I have some comments (some are listed here and others in the manuscript):

 Title:

-          It is very long and needs to be shorter as you mentioned (in the antimicrobial stewardship program) while you are evaluating the test by comparing it with the routinely used one (this is optional to the authors).

Abstract:

-          Subtitles (background, aim, results, and conclusion) should be added to be clear.

Introduction:

-          In lines 40, and 41, (The estimated mortality rate due to sepsis is around 30%) you need to mention the year for this estimation and if it is local or global.

-          Some facts are mentioned without citing the reference (lines 54 to 62) (the comment in the PFD).

Methods:

-          Please include the inclusion and exclusion criteria for each group.

-          Please cite your references for the methods used in blood culture & Vitek.

Best regards,

Author Response

The authors would like to thank the Reviewers for their specific and helpful comments on the manuscript. The authors have carefully considered the comments and have made revisions to the manuscript to address the reviewer’s concerns.

Responses to the Reviewer are reported as follows:

Reviewer 2:

In the current manuscript, you have analyzed the accuracy and time efficiency of the Film Array blood culture identification (FA-BCID) panel in identifying the pathogens in positive blood cultures via testing the results of 272 individuals (212 controls, and 60 FA-BCID participants. Results showed that 73% (44/60) of the blood samples demonstrated agreement between FA-BCID and MALDI-TOF. The FA-BCID panel detected mecA genes in seven Staphylococcus species; six cases were confirmed by antimicrobial susceptibility testing. In addition, KPC genes were detected in one Escherichia coli and one Klebsiella pneumoniae, although only the latter corresponded with the result from antimicrobial susceptibility tests. The turnaround time (TAT) for identification by FA-BCID was shorter. They suggested that the FA-BCID panel provides an identification result that is as reliable as that provided by the routine identification procedure but with shorter TAT; thus, the FA-BCID method is considered an effective and beneficial method for therapeutic decision-making and the improvement of the ASP for patients with bloodstream infection.

The work is very good and interesting. However, I have some comments (some are listed here and others in the manuscript):

  1. Title:

It is very long and needs to be shorter as you mentioned (in the antimicrobial stewardship program) while you are evaluating the test by comparing it with the routinely used one (this is optional to the authors).

Response:

We altered the title according to the suggestion from the reviewer.

“The significance of FilmArray blood culture identification panel (FA-BCID) for managing patients with positive blood culture.”

  1. Abstract:

-          Subtitles (background, aim, results, and conclusion) should be added to be precise.

Response:

The abstract was written as per the instructions from the journal; hence we did not write it structurally (with subtitles of background, aim, results, and conclusion), but it was written as one paragraph (unstructured).

Author instruction, Manuscript preparation

“An unstructured abstract of no more than 200 words should be included as well as a list of 3 to 10 keywords.”

Introduction:

  1. In lines 40, and 41, (The estimated mortality rate due to sepsis is around 30%) you need to mention the year for this estimation and if it is local or global.

Response:

As suggested by the reviewer, we have added the information required.

Lines: 40-41

“The estimated mortality rate due to sepsis in Korea from 2012 to 2022 is around 30%”

  1. Some facts are mentioned without citing the reference (lines 54 to 62) (the comment in the PFD).

Response:

We have added the reference as suggested by the reviewer.

Lines: 60, 63

“The additional assessment in conventional blood culture identification undoubtedly adds to the time before the results are available [9]. In addition, traditional methods are relatively labor-intensive and require expertise for evaluation; thus, alternative identification methods that are somewhat less labor-intensive yet enable rapid detection are preferable for clinical diagnosis [8, 9].”

Methods:

  1. Please include the inclusion and exclusion criteria for each group.

Response:

As suggested by the reviewer, we have added the inclusion and exclusion of the study.

Lines: 84-88

“Samples obtained from patients admitted to the emergency department in GNUCH that were suspected of a positive blood culture were included in the study. The samples from patients who died between the blood culture sampling and positivity detection were excluded from the study.”

  1. Please cite your references for the methods used in blood culture & Vitek.

Response:

As suggested by the reviewer we have added the references for the methods used in blood culture and AST (vitek).

“Briefly, for each isolate, the suspensions were prepared and used to inoculate the AST card. The results were read and interpreted automatically using software (VITEK® 2 Advanced Expert System, version 9.02) provided by bioMérieux [16].”

Lines: 105, 139-140